# Growth behavior and glyphosate resistance level in 10 populations of *Echinochloa colona* in Australia

**Gulshan Mahajan** [ID]*, **Vishavdeep Kaur, Michael Thompson, Bhagirath Singh Chauhan**

Queensland Alliance for Agriculture and Food Innovation (QAAFI), The University of Queensland, Gatton, Australia

* g.mahajan@uq.edu.au

**Data Availability Statement:** All relevant data are within the paper and its Supporting Information files.

## Abstract

Recently, poor control of *Echinochloa colona* with glyphosate has been reported in no-till agriculture systems of the northern grain region (NGR) of Australia. Two experiments were conducted using 10 populations of *E. colona* selected from the NGR of Australia to understand differences in their growth behavior and resistance pattern. Growth studies revealed that these populations differed in plant height (53–70 cm plant$^{-1}$), tiller production (30–52 tillers plant$^{-1}$), leaf production (124–186 leaves plant$^{-1}$) and seed head production (37–65 seed heads plant$^{-1}$). Days taken to seed heads and shoot biomass in these populations ranged between 40–48 d and 21–27 g plant$^{-1}$, respectively. Seed production in these populations ranged between 5380 and 10244 seeds plant$^{-1}$; lowest for population B17/25 and highest for population B17/13. Correlation studies revealed that seed number plant$^{-1}$ had a positive correlation with tiller number plant$^{-1}$ ($r = 0.73$) and negative relation with days taken to seed head initiation ($r = -0.65$). The glyphosate dose-response study showed a wide range of responses in these populations and the glyphosate dose required to kill 50% plants (LD$_{50}$ values) was estimated between 161 to 2339 g a.e. glyphosate ha$^{-1}$. LD$_{50}$ values of populations B17/16, B 17/34 and B17/35 were 1086, 2339 and 1153 g ha$^{-1}$, respectively, making them 6.7, 15.1 and 7.2-fold resistant to glyphosate compared with the susceptible population B17/37. Growth behavior and seed production potential in these populations had no correlation with the resistance index. These results suggest that some populations of *E. colona* are highly problematic; for example, population B17/34 was not only highly glyphosate-resistant, but also produced a high seed number (9300 seeds plant$^{-1}$). This study demonstrated that there is a possibility of great risk with the increased use of glyphosate for managing *E. colona* in the NGR of Australia. The results warrant integrated weed management strategies and improved stewardship guidelines are required for managing glyphosate-resistant populations of *E. colona* and to restrict further movement of resistant populations to other regions of Australia.

**Funding:** The authors (s) received no specific funding for the work.

**Competing interests:** The authors have declared that no competing interests exist.

## Introduction

*Echinochloa colona* (L.) Link ($C_4$ plant) has emerged as a major weed in summer crops in Australia and competes highly for water, sunlight and nutrients [1, 2]. *E. colona* is widely distributed in the northern grain region (NGR) of Australia [3, 4, 5] and it costs Australian agriculture AU$ 14.7 million annually [6]. Therefore, it affects the economy of Australian agriculture enormously. Emergence of multiple cohorts in the summer season, along with high capacity for seed production and seed dispersal have allowed the spread of *E. colona* throughout the NGR of Australia. The seeds remain viable in the soil for more than one year, causing continuous recruitment [7]. In Australia, intraspecific variations in *E. colona* have been reported on the basis of genetic diversity [8]. Morphological studies of these populations may increase our knowledge further and identify how these populations adapt to climate change and play a role in invasiveness. A minor change in morphology or physiology of the plant may affect its adaptability in a changing climate and a large number of dispersed seeds in the field, combined with the ability of this weed to flower under a range of photoperiods, may contribute to its invasiveness [9].

In the NGR of Australia, *E. colona* is a very common weed in no-till fallow land and glyphosate spray is the most common management practice for managing this weed. Glyphosate was mostly used in orchards (high-value crops) when introduced in Australia during the 1970s, as it was relatively expensive [10]. However, in the 1980s, its price declined, and its application became a common practice for weed control in a pre-seeding and fallow situation in Australia, which enabled the growers to adopt the conservation tillage practice. Glyphosate disrupts the shikimate pathways, reducing aromatic acid production via inhibition of the chloroplast enzyme, 5-enolpyruvylshikimate-3-phosphate synthase (EPSPS). Presently, control of some populations of *E. colona* in the NGR has become difficult with glyphosate as it has evolved resistance. The first case of glyphosate-resistant *E. colona* was reported in the NGR in 2007 [11]. At present, 41 weed species have been reported as glyphosate-resistant worldwide [12]. The evolved resistance may be due to intensive and repeated use of glyphosate [13, 14]. *E. colona* has also evolved resistance to four other herbicide modes of action, in addition to glyphosate [15].

A better understanding of the differences between populations for control with glyphosate is essential for developing long-term strategies. Variation in growth, morphological and physiological characteristics may alter herbicide efficacy within a species. Efficacy of glyphosate can be affected by plant species, population, plant development stage and environmental conditions [16]. Further, herbicide-resistant populations can spread from one area to another through pollen, seed or other propagules [17, 18]. Therefore, it is important to understand characteristics of resistant populations of a specific area to make better decisions and long-term strategies for weed control [19, 20]. A dose-response experiment is often conducted to assess the level of resistance in different populations. The dose-response experiment identifies a dose of an herbicide that provides a 50% reduction in shoot biomass [21].

In the NGR of Australia, there is variability in control of *E. colona* with glyphosate. We hypothesized that the dose required to kill 50% plants ($LD_{50}$ value) may vary between populations due to development of different levels of glyphosate resistance. It was also hypothesized that the reproduction potential of these populations may differ due to variability in the resistance factor. Information on resistant factor, and growth and reproduction behavior of these populations is limited in the NGR of Australia. Keeping these points in view, this study was planned to evaluate the growth, reproduction behavior and level of glyphosate resistance in different populations of *E. colona*. In this study, one experiment evaluated the growth and

reproduction behavior of 10 populations of *E. colo*na from the NGR of Australia and another experiment evaluated the sensitivity of these populations to glyphosate.

## Results and discussion

### Growth and seed production

Analysis of variance (ANOVA) for various parameters of *E. colona* populations has been presented in Table 1. Amongst populations, the final plant height ranged from 53 to 70 cm, where the lowest was B17/35 and highest was B17/16. Populations B17/16, B17/17 and B17/25 attained a similar height, however, they were taller than populations B17/34 and B17/35 (Table 2). Tiller number among different populations ranged between 30 to 52 plant$^{-1}$, where the lowest was B17/25 and highest was B17/49 (Table 2). Populations B17/7, B17/12, B17/13 and B17/49 produced similar tiller numbers plant$^{-1}$, however, their tiller production was higher than populations B17/25 and B17/35. Leaf numbers in different populations varied from 124 to 192 leaves plant$^{-1}$, where the lowest was population B17/16 and highest was population B17/34. Leaf production (numbers plant$^{-1}$) remained similar for populations B17/34, B17/35 and B17/49, however, leaf production in these populations was higher than populations B17/16 and B17/25. All populations produced similar numbers of seed heads except for B17/25, which produced lower numbers than the other populations (Table 2).

The weight of seed heads among different populations varied from 6.2 to 9.9 g plant$^{-1}$. It was similar for populations B17/7, B17/12, B17/25, B17/34, B17/35, and B17/37 (6.2 to 7.9 g plant$^{-1}$), however, these populations had a lower seed head weight than populations B17/16 (9.8 g plant$^{-1}$) and B17/49 (9.9 g plant$^{-1}$). Shoot biomass among different populations ranged between 20.9 to 27.3 g plant$^{-1}$ (Table 2). Shoot biomass remained similar for populations B17/13, B17/16 and B17/49, however, in these three populations, shoot biomass was significantly higher than populations B17/34, B17/35 and B17/37. Root biomass did not vary among populations (Table 2).

Time taken to seed head initiation in different populations varied from 40 to 48 d. Populations B17/7, B17/12, B17/13, and B17/17 took a similar time for seed head initiation (40–42 d) and produced seed heads earlier than populations B17/16, B17/25 and B17/35, of which B17/35 took the longest (48 d). Seed production in different populations varied from 5380 to 10244 seeds plant$^{-1}$; where the lowest was population B17/25 and highest was population B17/13. Populations B17/12, B17/13, B17/34 and B17/49 produced a similar number of seeds (8298–10244 plant$^{-1}$), with their seed production being higher than populations B17/25 and B17/35.

A linear positive correlation was found for seed number with tiller number plant$^{-1}$ ($r$ = 0.73) (Table 3). Tiller number plant$^{-1}$ had a positive correlation with seed heads plant$^{-1}$ ($r$ = 0.66). Seed head plant$^{-1}$ had a negative relation with shoot biomass plant$^{-1}$ ($r$ = - 0.73); however, relation of seed head plant$^{-1}$ with leaf number plant$^{-1}$ was positive ($r$ = 0.67). Shoot biomass had a positive relation with seed head weight ($r$ = 0.76). (Table 3). Plant height had a negative relation with leaf number plant$^{-1}$ ($r$ = - 0.91). Seed number plant$^{-1}$ had a negative relation with days taken to seed head initiation ($r$ = - 0.65).

The results of this study demonstrated that characteristics like high tillering capacity allow *E. colona* populations to produce a high leaf number that resulted in a large number of seed heads and seeds. Therefore, there is a need to restrict high tiller production in *E. colona* to reduce seed numbers. A recent study on crop-weed interference suggested that crop competition could reduce tiller numbers in *E. colona* [22]. In Australia, farmers are following wide and skip row spacing in crops such as cotton (*Gossypium hirsutum* L.), mungbean [*Vigna radiata* (L.) R. Wilczek] and sorghum [*Sorghum bicolor* (L.) Moench]; therefore, wide space between the rows could provide a better opportunity to *E. colona* populations with a high tillering

**Table 1. Analyses of variance for various parameters in different populations of *Echinochloa colona*.**

| Source | Degree of freedom (df) | Plant height (cm) | Tiller (number plant⁻¹) | Leaf (number plant⁻¹) | Seed head (number plant⁻¹) | Seed head weight (g plant⁻¹) | Shoot biomass (g plant⁻¹) | Root biomass (g plant⁻¹) | Days to seed head initiation (d) | Seed production (number plant⁻¹) |
|---|---|---|---|---|---|---|---|---|---|---|
| Replication | 7 | 24850 | 1425 | 9852 | 1347 | 29.7 | 2205.9 | 57.5 | 37.3 | 45900730 |
| Treatment | 9 | 203 | 323 | 4340 | 582 | 12.5 | 39.8 | 50.1 | 43.0 | 18826700 |
| Error | 63 | 35 | 48 | 932 | 181 | 2.79 | 16.3 | 27.9 | 9.5 | 4701360 |

capacity nature as compared to when crops are sown in narrow rows. In these environments (wide rows and fallows), *E. colona* could attain its high tillering potential and could produce a high seed number. *E. colona* in the present study produced tillers in the range of 39 to 52 plant⁻¹; however, in a previous study conducted in Greece, it produced tillers in the range of 115 to 131 plant⁻¹ [23]. This difference could be due to genotype x environment interactions and differential pot size. In the present study, we observed that populations B17/13 and B17/49 had higher tillers than populations B17/25 and B 17/35. This also suggested that genotypes and environmental interactions played a role in influencing tiller numbers per plant in *E. colona* populations. The regions where populations are of high tillering capacity are expected to suffer a high crop yield loss due to high *E. colona* competition.

The high seed number observed in populations B17/12, B17/13, B17/34 and B17/49 was largely attributed to a high number of tillers. The number of leaves and seed heads were similar between populations B17/13 and B17/35; however, seed production was lower in B17/35, which could be due to the lower tiller production and seed head weight in B17/35. The time taken to seed head initiation in the present study was similar to a study conducted in northern Greece, in which *E. colona* attained seed heads between 39 to 45 days after transplanting [23]. The population B17/35 (selected from the Moree region) took a longer time for seed head initiation than other populations (Fig 1). In a previous study in South-East Asia, 12 *E. colona* populations were studied and it was found that time for seed heads in different populations varied with latitude and plants from a high latitude attained seed heads earlier than from a low latitude [24]. This suggested that growth duration in different populations of *E. colona* may vary with geographical location. In the present study, the negative relationship between seed head

**Table 2. Morphological traits and seed production potential of different populations of *Echinochloa colona* (mean ± standard error of eight replicates).**

| Population | Plant height (cm) | Tiller (number plant⁻¹) | Leaf (number plant⁻¹) | Seed head (number plant⁻¹) | Seed head weight (g plant⁻¹) | Shoot biomass (g plant⁻¹) | Root biomass (g plant⁻¹) | Days to seed head initiation (d) | Seed production (number plant⁻¹) |
|---|---|---|---|---|---|---|---|---|---|
| B17/7 | 62.9 ± 5.8 | 47 ± 6.2 | 135 ± 16 | 53 ± 10 | 7.7 ± 1.0 | 24.8 ± 5.9 | 12.7 ± 0.8 | 42 ± 1.8 | 7022 ± 1245 |
| B17/12 | 61.2 ± 4.9 | 47 ± 5.9 | 164 ± 15 | 64 ± 7 | 7.5 ± 0.87 | 24.2 ± 5.4 | 13.0 ± 1.9 | 41 ± 0.9 | 8837 ± 1245 |
| B17/13 | 63.7 ± 4.1 | 50 ± 4.2 | 156 ± 11 | 59 ± 4 | 8.5 ± 0.69 | 25.1 ± 5.3 | 18.0 ± 2.6 | 40 ± 0.8 | 10244 ± 1676 |
| B17/16 | 70.3 ± 7.2 | 40 ± 4.6 | 124 ± 11 | 53 ± 3 | 9.8 ± 0.92 | 27.3 ± 6.5 | 10.1 ± 1.5 | 45 ± 1.1 | 6986 ± 895 |
| B17/17 | 66.9 ± 6.9 | 41 ± 5.0 | 134 ± 17 | 60 ± 7 | 8.1 ± 0.80 | 22.7 ± 5.5 | 9.7 ± 0.8 | 42 ± 0.6 | 7801 ± 1004 |
| B17/25 | 66.7 ± 5.8 | 30 ± 3.8 | 132 ± 10 | 37 ± 4 | 6.2 ± 0.81 | 22.1 ± 5.3 | 13.7 ± 1.9 | 45 ± 1.4 | 5380 ± 729 |
| B17/34 | 56.2 ± 5.2 | 43 ± 3.9 | 192 ± 14 | 63 ± 5 | 7.9 ± 0.66 | 20.9 ± 4.3 | 12.5 ± 1.3 | 44 ± 1.8 | 9295 ± 892 |
| B17/35 | 53.1 ± 6.1 | 39 ± 4.3 | 186 ± 18 | 65 ± 6 | 6.2 ± 0.60 | 20.9 ± 5.1 | 16.1 ± 3.4 | 48 ± 1.7 | 6130 ± 893 |
| B17/37 | 63.2 ± 6.3 | 43 ± 5.1 | 162 ± 7 | 59 ± 5 | 7.5 ± 0.91 | 21.1 ± 4.5 | 12.4 ± 2.3 | 43 ± 0.7 | 6387 ± 768 |
| B17/49 | 62.9 ± 6.2 | 52 ± 4.7 | 166 ± 18 | 64 ± 6 | 9.9 ± 0.88 | 25.4 ± 5.8 | 11.9 ± 1.5 | 44 ± 0.7 | 8298 ± 1214 |
| LSD (0.05) | 5.7 | 7.0 | 30.5 | 14.2 | 1.7 | 3.8 | NS | 2.7 | 2136 |

NS: nonsignificant.

**Table 3. Correlation of morphological traits with seed number and R/S factor in different populations of *Echinochloa colona* (n = 10).**

| Parameter | Plant height (cm) | Tiller (number plant⁻¹) | Leaf (number plant⁻¹) | Seed head (number plant⁻¹) | Seed head weight (g) | Shoot biomass (g plant⁻¹) | Root biomass (g plant⁻¹) | Days to seed head initiation (d) | Seed production (number plant⁻¹) | Resistant index |
|---|---|---|---|---|---|---|---|---|---|---|
| Tiller (number plant⁻¹) | -0.14 | | | | | | | | | |
| Leaf (number plant⁻¹) | -0.91* | 0.28 | | | | | | | | |
| Seed head (number plant⁻¹) | -0.56 | 0.66* | 0.67* | | | | | | | |
| Seed head weight (g plant⁻¹) | 0.46 | 0.60 | -0.21 | 0.28 | | | | | | |
| Shoot biomass (g plant⁻¹) | 0.61 | 0.44 | -0.56 | -0.73* | 0.76* | | | | | |
| Root biomass (g plant⁻¹) | -0.49 | 0.12 | 0.41 | 0.058 | -0.41 | -0.18 | | | | |
| Days to seed head initiation (d) | -0.34 | -0.57 | 0.23 | -0.100 | -0.28 | -0.33 | -0.037 | | | |
| Seed production (number plant⁻¹) | -0.13 | 0.73* | 0.33 | 0.56 | 0.47 | 0.28 | 0.22 | -0.65* | | |
| Resistant index | -0.50 | -0.10 | 0.56 | 0.29 | 0.065 | -0.26 | -0.017 | 0.44 | 0.23 | |

Critical value of *r* at 5% = 0.63;

* indicates significant relation.

initiation and seed number revealed that late-maturing populations produced fewer seeds as was the case for populations B17/16 and B17/35 when compared with population B17/13. These results suggest that diversity in *E. colona* traits could result in differential responses to herbicides, cultural practices, and resistance evolution. For example, the early vigor trait in *E. colona* is an important trait that could affect early crop-weed competition [25] and therefore, management of such populations at an early stage is required to increase crop production and reduce the weed seed bank in the soil.

In this study, *E. colona* populations differed in their seed production potential, which ranged between 5380 to 10240 seeds plant⁻¹. Differential seed production in *E. colona* populations could play an effective role in its spread and population establishment [26]. High seed yields in populations B17/12, B17/13, B17/34 and B17/49 were largely based on a greater number of seed heads and leaf numbers plant⁻¹. High production of leaves in these populations probably maintained a better supply rate of carbon assimilates to seeds. In one study on *Brassica*, it was found that variation in the supply of carbon assimilates to seeds at or immediately after anthesis could cause a variation in seed production in different populations [27]. Some authors also highlighted the role of the supply of carbon assimilates in determining the seed number in pea (*Pisum sativum* L.) plant [28].

The present study also revealed that tiller number per plant played a large role in seed production along with leaf number per plant. Population B17/35 had high leaf production but could not produce higher amounts of seeds like B17/13 and B17/34 did, because it had lower tiller production than B17/13 and B17/34. Although this study revealed that the supply of carbon assimilates after anthesis could be a major factor in determining seed production, we

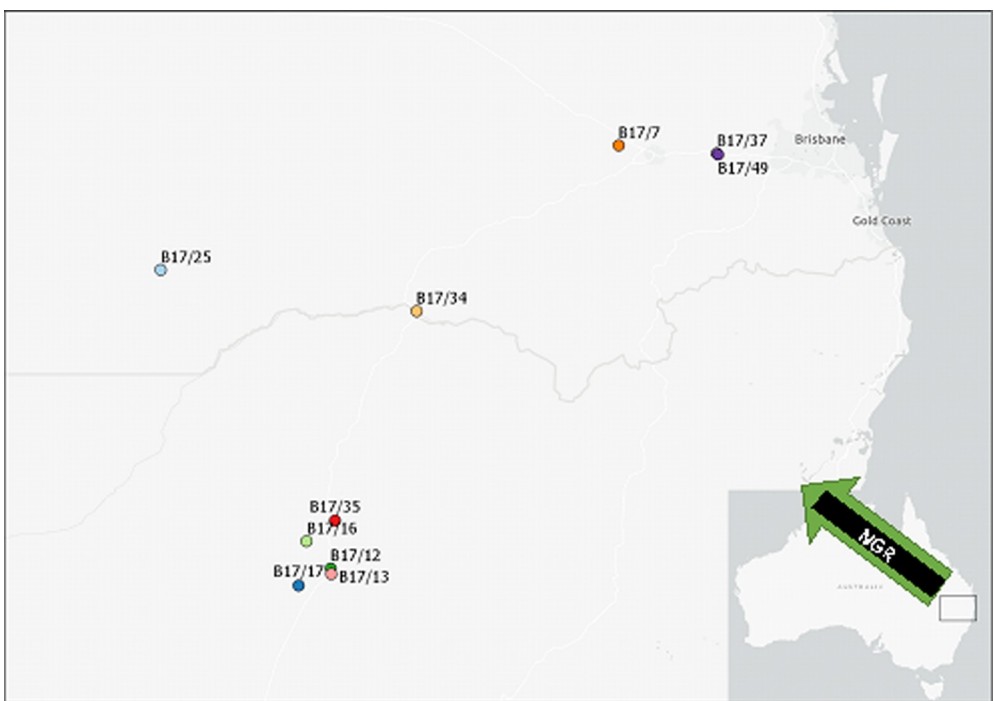

**Fig 1. Location of 10 populations of *Echinochloa colona* selected from the northern grain region (NGR) of Australia.**

could not rule out the possibility of hormonal factors for variation in seed production in these populations. These results suggest that there is also a need to study nutritional and hormonal factors for variation in seed production in these populations [27]. Our study (second experiment) also found that the $LD_{50}$ value of glyphosate for these populations varied. These results suggest that in these populations, seed viability, seed persistence and fitness penalty may differ and therefore systematic studies need to be investigated. Such knowledge of seed production in these populations is required for understanding the evolution and spread of herbicide resistance particularly for herbicide-resistant populations.

**Table 4. Probit transformed response [Intercept + b*x* (covariate *x* are transformed using the base 10.0 logarithm)] for different *Echinochloa colona* populations, \*represents significant (P<0.05).**

| Population | *Response* | $R^2$ | Pearson Goodness- of-fit *Chi square* | Significance level (*Chi square*) | 95% confidence interval | |
|---|---|---|---|---|---|---|
| | | | | | Lower bound | Upper bound |
| B17/7 | $y = -6.92 + 2.99\,x$ | 0.98 | 0.638 | 0.73* | 151.0 | 275.3 |
| B17/12 | $y = -4.78 + 2.09\,x$ | 0.94 | 2.421 | 0.30* | 142.9 | 282.2 |
| B17/13 | $y = -6.27 + 2.45\,x$ | 0.97 | 3.560 | 0.17* | 302.1 | 436.9 |
| B17/16 | $y = -4.52 + 1.49\,x$ | 0.99 | 0.663 | 0.72* | 884.4 | 1356.5 |
| B17/17 | $y = -5.79 + 2.52\,x$ | 1.00 | 1.864 | 0.39* | 157.5 | 283.3 |
| B17/25 | $y = -3.61 + 1.49\,x$ | 0.93 | 3.019 | 0.22* | 129.3 | 342.1 |
| B17/34 | $y = -4.96 + 1.47\,x$ | 0.98 | 1.249 | 0.54* | 1801.2 | 3500.3 |
| B17/35 | $y = -4.43 + 1.45\,x$ | 0.98 | 1.185 | 0.55* | 937.7 | 1461.3 |
| B17/37 | $y = -2.83 + 1.46\,x$ | 1.00 | 3.659 | 0.16* | 76.8 | 227.4 |
| B17/49 | $y = -4.74 + 1.77\,x$ | 0.99 | 0.562 | 0.58* | 375.2 | 583.0 |

## Response to glyphosate

Out of 10 *E. colona* populations collected from the NGR of Australia, three populations (B17/16, B 17/34 and B17/35) had greater than 80% survival following treatments with 325 to 2600 g a.e. ha$^{-1}$ glyphosate. The probit analysis details for each population along with their level of significance is presented in Table 4. The dose-response study of glyphosate for these populations showed a wide range of responses (Fig 2A). The $LD_{50}$ value of the tested populations ranged from 161 to 2339 g ha$^{-1}$ (Fig 2A). The susceptible population B17/37 was easily controlled with glyphosate and had a $LD_{50}$ of 161 g ha$^{-1}$, below the normal use rate of this herbicide (650 g ha$^{-1}$). The $LD_{50}$ values of populations B17/16, B17/34 and B17/35 were 1086, 2229 and 1153 g ha$^{-1}$, respectively, making them 6.7, 15.1 and 7.2-fold resistant to glyphosate compared with the susceptible population B17/37 (Fig 2B). Correlation studies revealed that growth behavior and seed production potential in these populations had no correlation with the resistance index (Table 4). The most resistant population B17/34 was from the Goondiwindi region, whereas the next most resistant populations, B17/35 and B17/16, were from the Moree and Narrabri regions, respectively. This study has revealed that *E. colona* populations in the NGR of Australia have different levels of resistance to glyphosate. No-till farming is quite popular in the NGR of Australia for moisture conservation. Therefore, growers use glyphosate in summer fallows to kill weeds and conserve moisture. Repeated and intensive use of glyphosate in this region has evolved glyphosate resistance in these populations [11]. Risk of glyphosate resistance evolution for weeds is higher in those areas where glyphosate has been used for a long time and with minimal soil disturbance [29]. In Australia, glyphosate-tolerant cotton varieties are very popular among cotton growers. Glyphosate-resistant *E. colona* populations may create serious situations in that production environment. The resistant factor for glyphosate in this study was similar to the first reported case of glyphosate resistance in Australia (7 to 11-fold resistance compared with a susceptible population); but that study was reported for rigid ryegrass (*Lolium rigidum* Gaud.) [30].

Earlier, glyphosate resistance in *E. colona* populations was also reported in Australia [11, 31]. There are a number of mechanisms responsible for glyphosate resistance [32], and different mechanisms may result in a different level of resistance [33]. Therefore, these studies suggests that these resistant populations may not carry the same resistance allele, which needs to be investigated. Many reports of glyphosate resistance in different weeds highlight that the reliance on glyphosate for weed control, in the long run, exerts a substantial selection pressure on weeds [34,35,36,37,38,39,40]. Therefore, integrated weed control should be strengthened to reduce selection pressure on these resistant populations, particularly in cotton paddocks. It is quite possible that the mechanism of glyphosate resistance in Australian *E. colona* populations might be different from resistant *E. colona* populations reported from California [41] as Australian populations of *E. colona* have adapted to a dry environment. Therefore, a systematic study is required to understand the evolution of glyphosate resistance in these populations. The evolution of glyphosate resistance in tropical *E. colona* in Australia suggests that there is a large risk with increased use of glyphosate in fallows and improved stewardship guidelines for glyphosate use are required in the NGR of Australia.

## Conclusions

The present study on *E. colona* populations has increased our understanding of the physiological basis of differences in seed production due to variations in morphological characteristics and resistance behavior. It highlighted that growth parameters such as high tiller production in *E. colona* populations leads to more seed heads and in turn high seed production. The study further demonstrated that growth behavior and seed production potential in these populations

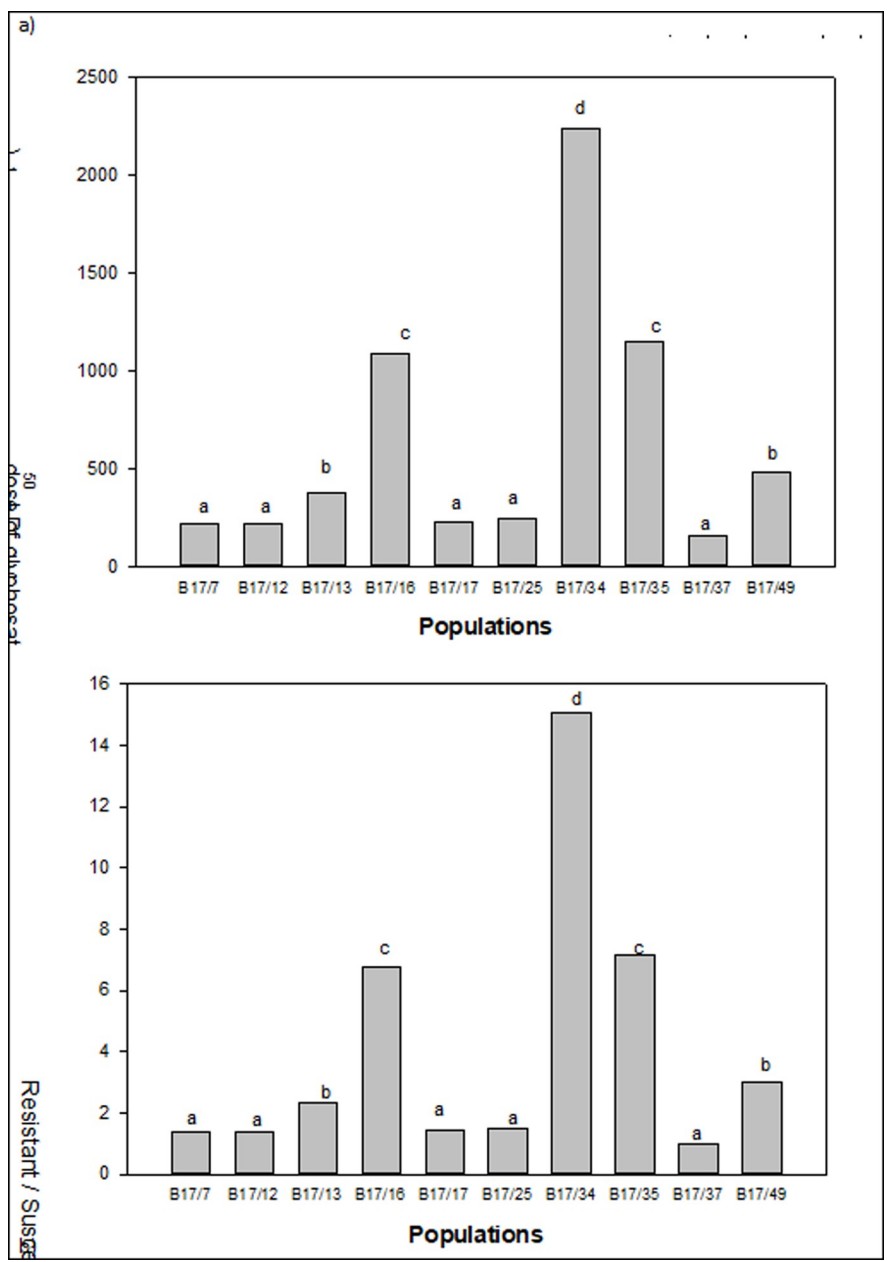

**Fig 2.  a)** $LD_{50}$ dose of glyphosate and **b)** glyphosate-resistant/susceptible factor in different populations of *Echinochloa colona* (bars followed by same letter are not significantly different on the basis of 95% confidence interval).

had no correlation with the resistance index. However, this research has posed more questions than it has answered. This study suggested that populations such as B17/34 that are highly glyphosate-resistant, and also produced a high seed number (9300 seeds plant$^{-1}$) are very problematic. Therefore, systematic research on weed biology, physiology and resistance mechanism is required to answer these questions for better understanding. Efforts need to be made to reduce the invasion of herbicide-resistant populations, such as B17/34. Movement of resistant populations (seeds) from one field to another field via different modes of transportation should be restricted. For the management of resistant populations, crop competition tactics (use of narrow row spacing, competitive cultivars, optimum seeding rates, cover crops, etc.)

could be explored in an integrated weed management program to reduce the seed number/ seedling recruitment of resistant populations. This study also suggested that there is a need to understand the likelihood of resistance transfer from resistant to susceptible populations through pollen-mediated gene flow and introgression. Such knowledge could be useful in restricting the further spread of glyphosate-resistant populations of *E. colona*.

## Materials and methods

The study was conducted at the QAAFI weed science laboratory and screen house of the University of Queensland, Gatton, Australia. Seeds of 10 different populations of *E. colona* were collected from the NGR of Australia in March 2017. The coordinates and details of these populations are depicted in Fig 1 and Table 5 respectively. For collection of seeds, permission was obtained from the consultants through phone calls and personal meetings. Seeds were collected from 40–50 plants per site and over an area of >1ha. Seeds were shaken loose from the plants by hand for collection. Only matured plants were chosen for collection of seeds. For each population, about 10, 000 seeds were collected. Germination rate was tested to confirm seed viability and was found to be >80% for each population in both runs. The collected populations were designated on the basis of year and serial number in which these were collected, for example susceptible population B 17/37 indicates sample was collected in 2017 and 37 was the serial number of that population in the collection, while B stands for population. The seeds of each population were cleaned and stored in shade.

### Growth response experiment

In this experiment, 10 populations of *E. colona* were grown in pots replicated four times. The pots were kept on benches placed outside the screen house. Pots were filled with potting mix (Crasti & Company Pty Ltd, Sydney, Australia). Initially, 10 seeds were sown per pot at 1 cm depth and after establishment, one plant per pot was maintained. The experiment was conducted twice. The first run was started on 27 September 2018 and harvested on 6 December 2018. The second run was started on 3 December and harvested on 5 February 2019. Pots used in the experiment were 20 cm in height and arranged in a completely randomized design. The pots were regularly irrigated.

At maturity, plant height was measured from the base of the plant to the tip of the uppermost leaf of the plant. Days taken to seed head initiation was recorded in each pot. For estimating seed production per head, two intact seed heads were chosen randomly from each plant. For the total number of seeds, each rachilla segment (pedicel base) was counted and then, averaged for seeds per head. At harvesting time, tiller numbers, leaf numbers and seed heads per plant were also counted. Harvesting was done when ~80% seed heads matured.

At harvest, seed heads were separated from the plants for measuring shoot biomass. After that, all aboveground shoot biomass from each plant was placed separately in a paper bag and dried in an oven at 70°C for 72 hours before being weighed. For root weight data, pots containing potting mix with the root system were first dried in an oven at 70°C for 72 hours. After that, roots were removed from each pot by shaking loose the potting mix. Root biomass of each plant was then measured. Drying of potting mix in an oven helped in the separation of the root system from the potting mix.

### Glyphosate dose-response experiment

Seeds of 10 populations including glyphosate susceptible population B 17/37 were sown in pots (9 cm diameter and 10 cm height) filled with potting mix (Crasti & Company Pty Ltd, Sydney, Australia). Initially, 10 seeds were sown per pot at 1 cm depth and after establishment,

**Table 5. Detail of 10 populations of *E. colona* collected from the northern grain region of Australia.**

| Population | Coordinates | Location/Place | Crop |
|---|---|---|---|
| B17/7 | 27.50000000/151.69666667 | Dalby | Wheat-Fallow |
| B17/12 | 30.268508/149.80481 | Narrabri | Wheat- Fallow |
| B17/13 | 30.306499/149.811438 | Narrabri | Wheat-Fallow |
| B17/16 | 30.09099349/149.64890 | Narrabri | Lathyrus-wheat/chickpea |
| B17/17 | 30.38230/149.59679 | Narrabri | Fallow |
| B17/25 | 28.31500000/148.68916667 | St George | Canal side |
| B17/34 | 28.58305556/150.36888889 | Moree | Wheat fallow |
| B17/35 | 29.95805/149.8339 | Moree | Cotton fallow |
| B17/37 | 27.5514/152.3428 | Gatton | Wheat-Fallow |
| B17/49 | 27.5514/152.3428 | Gatton | Wheat-Fallow |

five plants per pot were maintained. Pots were kept in a screen house under natural light and temperature conditions. The experimental design was a factorial with four replicates where the first factor was population and the second factor was glyphosate dose [0x (no herbicide; control), 0.5x, 1x, 2x, and 4x]. The 1x dose was the recommended dose (650 g a.e. ha$^{-1}$) for glyphosate.

The commercial product of glyphosate named 'Glymount' containing active constituent 450 g/L was used and under fallow condition, its labelled rate for control of *E. colona* in Australia was 0.8–1.2 L ha$^{-1}$ (commercial dose).

The experiment was conducted twice. The first run was started on 5 December 2018 and harvested on 14 January 2019. The second run was started on 25 January 2019 and harvested on 6 March 2019. Glyphosate application was done on 24 December 2018 in the first run and 13 February 2019 in the second run. Plants were kept well-watered and fertilized.

Glyphosate was sprayed using a research track sprayer. Plants were treated at the 4–5 leaf stage using a spray volume of 108 L ha$^{-1}$ and Teejet XR 110015 flat fan nozzles were used. Plants were allowed to grow for 21 days after treatment (DAT) to determine glyphosate sensitivity. Plant survival was assessed 21 DAT, and plant aboveground biomass was harvested, dried for 72 hours at 70˚C, and weighed. As there were five seedlings for each population in four replications and the experiment was repeated twice; therefore, results were based on 40 seedlings per population. The susceptible population used in the study was B 17/37.

## Statistical analyses

The first experiment was conducted in a randomized completely block design and the second experiment was conducted in a randomized completely block design with a factorial arrangement. In both experiments, there was no interaction between experimental runs and treatments; therefore, the data of the two runs were pooled for ANOVA. All the data met assumptions of normality of residuals and homogeneity of variance. Data of the first experiment were subjected to analysis of variance (ANOVA) using the software Elementary Designs Application 1.0 Beta (AgriStudy.com: www.agristudy.com) (verified with GENSTAT 16th Edition; VSN International, Hemel Hempstead, UK). Treatment means were separated using Fisher's protected least significant differences (LSD) at P≤0.05. The relationships between parameters were assessed using linear regression analysis (see S1 Data).

For the second experiment, LD$_{50}$ (the dose required to kill 50%) estimates were generated using Probit analysis [IBM SPSS Statistics 20.0 (SPSS, Inc., Chicago, IL, USA)]. The level of significance was tested with a Chi-Square goodness of fit test (see S1 Text). When the calculated value of Chi-Square goodness of fit test was greater than the table value, the null hypothesis

was rejected and it was concluded that there was a significant difference between the observed and the expected value and vice versa with values lower than the table value. $LD_{50}$ values for each population were compared using confidence interval overlap. The resistance index (resistance/susceptibility ratio) was calculated on the basis of the $LD_{50}$ value to compare the resistance level among different populations.

## Supporting information

**S1 Data. Additional data and statistical analysis information.**
(XLSX)

**S1 Text. Probit analysis information.**
(DOCX)

## Author Contributions

**Conceptualization:** Gulshan Mahajan, Bhagirath Singh Chauhan.

**Data curation:** Gulshan Mahajan, Vishavdeep Kaur.

**Formal analysis:** Gulshan Mahajan.

**Methodology:** Gulshan Mahajan.

**Project administration:** Bhagirath Singh Chauhan.

**Resources:** Bhagirath Singh Chauhan.

**Supervision:** Bhagirath Singh Chauhan.

**Visualization:** Gulshan Mahajan.

**Writing – original draft:** Gulshan Mahajan.

**Writing – review & editing:** Gulshan Mahajan, Michael Thompson, Bhagirath Singh Chauhan.

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
