## [Decision Letter · Decision Letter 0]

9 Oct 2019

PONE-D-19-21773

Growth behavior and glyphosate resistance level in 10 biotypes of Echinochloa colona in Australia

PLOS ONE

Dear Dr Mahajan,

Thank you for submitting your manuscript to PLOS ONE. After careful consideration, we feel that it has merit but does not fully meet PLOS ONE’s publication criteria as it currently stands. Therefore, we invite you to submit a revised version of the manuscript that addresses the points raised during the review process.

ACADEMIC EDITOR: 

The comments of three reviewers have been received as presented below. Two of them recommended major revision while another one recommended rejection. Please revise the manuscript according to the comments of the three reviewers. Particularly, Rev.# 1 felt that the definition of herbicide resistance should be demonstrated according to reliable literature. Besides, two reviewers reviewed that statistical results have not been sufficiently presented and discussed. Please address all comments one by one and attach a response letter of a list or explanation of changes made to the manuscript in your revised version. 

We would appreciate receiving your revised manuscript by Nov 23 2019 11:59PM. To enhance the reproducibility of your results, we recommend that if applicable you deposit your laboratory protocols in protocols.io, where a protocol can be assigned its own identifier (DOI) such that it can be cited independently in the future. For instructions see: http://journals.plos.org/plosone/s/submission-guidelines#loc-laboratory-protocols

We look forward to receiving your revised manuscript.

Kind regards,

Xiao Guo, Ph.D.

Academic Editor

PLOS ONE

Journal Requirements:

1. In your Methods section, please provide additional information regarding the permits you obtained for the work. Please ensure you have included the full name of the authority that approved the collection sites access and, if no permits were required, a brief statement explaining why.

Reviewers' comments:

Reviewer's Responses to Questions

**Comments to the Author**

1. Is the manuscript technically sound, and do the data support the conclusions?

Reviewer #1: No

Reviewer #2: Yes

Reviewer #3: Yes

2. Has the statistical analysis been performed appropriately and rigorously? 

Reviewer #1: No

Reviewer #2: No

Reviewer #3: No

3. Have the authors made all data underlying the findings in their manuscript fully available?

Reviewer #1: No

Reviewer #2: No

Reviewer #3: Yes

4. Is the manuscript presented in an intelligible fashion and written in standard English?

Reviewer #1: Yes

Reviewer #2: Yes

Reviewer #3: Yes

5. Review Comments to the Author

Reviewer #1: Overall, the manuscript is written well and clearly. The subject should be of interest of readers of weed science community.

My biggest question is one of semantics, I believe. I read the criteria for listing a species as resistant on the website for International Survey of Herbicide Resistant Weeds (I. Heap; http://weedscience.com/Documents/ResistanceCriterion.pdf). There are problems with the research methods used that do not support the results presented. The definition of herbicide resistance is based on plant/population survival not plant growth (biomass). Check the WSSA website for the accepted definition of herbicide resistance. In the case of these studies where there was only few plants per pot, population survival cannot be effectively determined and therefore, it is not possible to definitively characterize the resistance status of the collected populations.

Once characterized, plant biomass data can be used to indicate the robustness of the resistance mechanism. However, this is only once resistance has been confirmed. The recommendation is that there is a need for further studies aimed at characterizing the resistance status of the collected biotypes based on survival of a reasonable population size (e.g. 40-50 seedlings). A known susceptible biotype needs to be included and replication is not needed if there is a sufficiently large population size. See methods in Owen et al 2016.

The whole plant dose response experiments appear to provide more definitive evidence of resistance. However, based on the author clarification of experimental design the dose response experiment may not be confirmatory if the study was not set up to statistically compare the putative resistant population with the susceptible population. Does the work also need to demonstrate heritability in order to confirm resistance in this case? I suspect literature evidence may be important since experimental evidence is lacking. Is there any plan to relate these phenotypic response back to management, cropping systems, location, and/or soils?

SPECIFIC COMMENTS

L 97: Please describe results and discussion in two separate sections.

L 251-254: How many plants per site were collected? What was the area occupied by those plants per location? Was there a protocol for collecting seed that defined the area of collection? What was the cropping and herbicide use history of each sampled field? How large a seed sample per population was obtained? Were sample sizes similar among populations? What was the germination percent for the sampled populations and did seed viability/germination rate stay constant for the two runs? The authors might consider including the designated names of collected biotypes.

L 285-287: How did you come up with 650 g/ha rate of glyphosate as 1X recommended rate? This is a low rate of glyphosate to go with. Who recommended this rate and for what situation? If this was based on chemical fallow, don’t you think 1260 g/ha rate will be much appropriate to go with. No adjuvant was included in those glyphosate doses? Dec 5, 2018 to Jan 14, 2018?

L 307-314: The section on statistical analyses is not complete with regard to the glyphosate dose-response analysis. How did you run the probit analysis? Provide the equation used and explained it here. How did you calculate the resistance index when there was no known susceptible biotype included?

Figure 2: Are those GR50 values or Resistant/susceptible factor significantly different from each other? If yes, What statistical test did you perform to find this out? Please explain in the figure legend.

I don’t see any depository for the raw data on these experiments. Please provide all your raw data from where you build these tables and figures. This is the most important requirement of the journal.

Reviewer #2: The topic is important. The whole paper is clear and well- structured. The experiment is reasonably designed. The result is mostly clear and fully described but I personally think that the result is probably not very well-dug. More analysis about the correlations among variables, the effect comparison of glyphosate usage could be conducted.

1. Please make sure with the journal whether the material and method parts are at the end of the paper or after introduction.

2. I might miss some information, but it is better having more descriptions about the codes of the species, for example, B17/34, does it have some relationship with the sampling (Fig.1)? It is not so clear and very easy to get confused with these codes when reading.

3. The Fig.1 seems not so clear and short of legend and north arrow. The color you chose is too similar with each other which makes the whole figure looks not so good.

4. There are correlations among leaf, tiller, amount of seeds, etc. Tables are good, but if you have several scatter plots to show these relationships, it might be clearer.

5. For the influences of height, leaf, etc. to the final tiller of the species, there might be interactive effects among different x variables. Is it better you use the multi- correlation analysis, i.e., y = ax1+bx2+cx3…? This is just a suggestion and you could think about this.

6. Line 134~135, it is not clear using “target” here.

7. I am not sure, for the unit, you use number plant-1 etc., do you think plant-1 here is proper? Is it better just delete it and make some explanations in the text? Because all unit is measured by one individual. Or you just use stem-1or individual-1?

8. In table 2, delete the last number “1”.

9. For table 3, make clear the meaning of each column at the title. “Probit analysis detail” is not clear.

10. Line 275, “by shaking” is not precise, you might find another word to describe this operation. Measures taken to reduce the root loss is better also addressed.

11. Line 305~306, more explanation might be needed for the “Fisher’s protected LSD”? At least give the full name of LSD when it is firstly mentioned.

12. Is the different doses of the glyphosate used well compared? It is not clear in the result.

13. For reference 6 and 7, there are italic styles, make sure with journal and be consist with the others.

14. You declared that the data is available in the supporting information but I did not see any materials attached, please explain this.

Reviewer #3: Two experiments were carried out in this study to compare the growth behavior and glyphosate resistance among 10 biotypes of Echinochloa colona in Australia. The growth experiment showed that characteristics like tall nature and high tillering capacity allow E. colona to produce a high leaf number that result in a large number of seed heads and seeds. The glyphosate dose-response experiment showed a wide range of GR50 value in these biotypes, and the biotype B17/34 showed the strongest resistance to glyphosate. The two experiments were conducted rigorously with appropriate replication and sample sizes, and the conclusions were drawn appropriately based on the data presented. In addition, the language of this manuscript is well written and clear.

The analysis of variance (ANOVA) and Fisher’s protected LSD were applied to test the difference of growth behavior of E. colona among 10 biotypes. And the probit anaylsis were applied to estimate the GR50 value. These statistical methods were performed appropriately, but the outcome of these statistical results have not been sufficiently presented. Furthermore, I suggest the multivariate analysis on the growth behavior and glyphosate resistance can be applied in this study to test the differences among 10 biotypes.

Moreover, there are several major and specific comments on this manuscript:

1. The introduction can be reduced to 3 or 4 paragraphs to make the background and aim clearer. The outline of the introduction can refer to:

1) The Echinochloa colona as a noxious weed emerged in Australia crops.

2) The biological characteristics of growth and reproduction of E. colona.

3) The glyphosate is widely used in Australia to eliminate the E. colona, but induced a variety of glyphosate resistant biotypes.

4）the aim and scientific questions of this study

2. The author combined the “results” and “discussion” in the manuscript. Although this format can be accepted by some journals, but I suggest divided this part into “results” and “discussions” separately. In the present manuscript, the result of two experiment were discussed respectively, and I can not find their relationships. If there is an independent section of discussion, the relationships between the two experiment and their implications can be better integrated.

3. The author proposed “growth behavior and seed production potential in these biotypes had no correlation with the resistance index” (LINE 240-241). However, the statistical methods for analyzing the correlation between growth traits and resistance index were not given in the section of “statistical analysis”. Also, I can not find the relevant result which support this conclusion in the context.

4. The seed of 10 biotypes of E. colona were collected from a large area in northern grain region in Australia. So, do the geographical differences effect on the growth behavior and glyphosate resistance among 10 biotypes of E. colona. I suggest the multivariate analysis such as PCA or RDA can be applied to study whether the biotypes from closer sites have similar characteristics. Moreover, the correlation (or non-correlation) between growth behavior and resistance index may also be revealed by the multivariate analysis.

Here are also several specific comments:

5. Figure 1. It’s better to give a small-scale map of Australia, and pointed out the location of northern grain region.

6. Table 1. the standard deviation of these variables should be given. And as the LSD was performed, I suggest the author add lowercase superscript letters after the values to indicate significance at the 0.05 level for the differences among biotypes.

7. LINE 64: “shikmate pathway” might be “shikimate pathway”

8. Line 251-254: it better to give more information on the seed collected location or the farm, like the crop type, soil type, glyphosate dose and so on, which can help readers know the growth environment of these biotypes.

9. Line 252: how to identify different biotypes? Do they just collected from different sites，or have distinct heritable characteristics?

10. Line 301: I did not find the ANOVA result

11. Line 307-308 Which variable did the “GR50” in this study based on? The shoot biomass, tiller number, or individuals?

6. PLOS authors have the option to publish the peer review history of their article (what does this mean?). If published, this will include your full peer review and any attached files.

Reviewer #1: No

Reviewer #2: Yes: Liping Li

Reviewer #3: No

---

## [Author Response · Author response to Decision Letter 0]

4 Nov 2019

ACADEMIC EDITOR:

The comments of three reviewers have been received as presented below. Two of them recommended major revision while another one recommended rejection. Please revise the manuscript according to the comments of the three reviewers. Particularly, Rev.# 1 felt that the definition of herbicide resistance should be demonstrated according to reliable literature. Besides, two reviewers reviewed that statistical results have not been sufficiently presented and discussed. Please address all comments one by one and attach a response letter of a list or explanation of changes made to the manuscript in your revised version. 

We have addressed all comments of three reviewers and attempt has been made to incorporate the critical comments of reviewer#1 carefully.

1. In your Methods section, please provide additional information regarding the permits you obtained for the work. Please ensure you have included the full name of the authority that approved the collection sites access and, if no permits were required, a brief statement explaining why.

Yes, permissions were obtained from the consultants through phones and personal meetings.

Reviewers' comments:

Reviewer #1: Overall, the manuscript is written well and clearly. The subject should be of interest of readers of weed science community.

My biggest question is one of semantics, I believe. I read the criteria for listing a species as resistant on the website for International Survey of Herbicide Resistant Weeds (I. Heap; http://weedscience.com/Documents/ResistanceCriterion.pdf). There are problems with the research methods used that do not support the results presented. The definition of herbicide resistance is based on plant/population survival not plant growth (biomass). Check the WSSA website for the accepted definition of herbicide resistance. In the case of these studies where there was only few plants per pot, population survival cannot be effectively determined and therefore, it is not possible to definitively characterize the resistance status of the collected populations.

We have also recorded the data for survival %. We again reanalysed the data and calculated LD50 value on the basis of survival %. We used several doses, so the number of plants used per pots were sufficient. Yes, for only one population, they were not enough.

Once characterized, plant biomass data can be used to indicate the robustness of the resistance mechanism. However, this is only once resistance has been confirmed. The recommendation is that there is a need for further studies aimed at characterizing the resistance status of the collected biotypes based on survival of a reasonable population size (e.g. 40-50 seedlings). A known susceptible biotype needs to be included and replication is not needed if there is a sufficiently large population size. See methods in Owen et al 2016.

There were five seedlings for each biotype in four replications. So 20 seedlings for each biotype. As the experiment was repeated twice; therefore, results are based on 40 seedlings per biotype. We had a susceptible biotype B 17/37. The data has been reanalysed for survival%.

The whole plant dose response experiments appear to provide more definitive evidence of resistance. However, based on the author clarification of experimental design the dose response experiment may not be confirmatory if the study was not set up to statistically compare the putative resistant population with the susceptible population. Does the work also need to demonstrate heritability in order to confirm resistance in this case? I suspect literature evidence may be important since experimental evidence is lacking. 

We followed the publication of Gaines et al. 2012 for experimental design and planning that was published in Weed Technology and they did not demonstrate heritability in order to confirm resistance. Yes, we had a susceptible biotype named B 17/37 in these 10 biotypes.

Gaines T.A., Cripps A., Powles S.B. Evolved resistance to glyphosate in junglerice (Echinochloa colona) from the tropical Ord River region in Australia. Weed Technology, 2012 26: p. 480-484.

SPECIFIC COMMENTS

L 97: Please describe results and discussion in two separate sections.

Results and discussion are combined here according to the format of the journal. Moreover, combined results and discussion is good to avoid repetition. However at the end, we have concluded our results in a general discussion.

L 251-254: How many plants per site were collected? What was the area occupied by those plants per location? Was there a protocol for collecting seed that defined the area of collection? How large a seed sample per population was obtained? Were sample sizes similar among populations? What was the germination percent for the sampled populations and did seed viability/germination rate stay constant for the two runs? The authors might consider including the designated names of collected biotypes. 

There were 40-50 plants per site and the area occupied by those plants per location was >1 ha. Seeds were collected directly from plants by shaking. Only matured plants were chosen for collection of seeds. There were about 10, 000 seeds per population. The sample size was different depending on infestation levels. The germination rate (seed viability) was >80% for each biotype in both the experimental runs. The collected biotypes were designated on the basis of year and serial number in which these were collected for example 17/37 was designated as sample was collected in 2017 and 37 was the serial number of that biotype in collection.

L 285-287: How did you come up with 650 g/ha rate of glyphosate as 1X recommended rate? This is a low rate of glyphosate to go with. Who recommended this rate and for what situation? If this was based on chemical fallow, don’t you think 1260 g/ha rate will be much appropriate to go with. No adjuvant was included in those glyphosate doses? Dec 5, 2018 to Jan 14, 2018?

The commercial product of glyphosate named Glymount containing active constituent 450 g/L was used and under fallow condition in Australia, its labelled rate is 0.8 -1.2 L/ha 

https://growchoice.com.au/wp-content/uploads/2015/05/Glymount-DFU.pdf.

Therefore, this is not a low rate of glyphosate in Australia. No adjuvant was included in glyphosate doses. We have corrected the date- it is Jan14, 2019.

L 307-314: The section on statistical analyses is not complete with regard to the glyphosate dose-response analysis. How did you run the probit analysis? Provide the equation used and explained it here. How did you calculate the resistance index when there was no known susceptible biotype included?

Probit analysis was done using software IBM SPSS Statistics 20.0 (SPSS, Inc., Chicago, IL, USA). The equation has been provided and explained. Resistance index was calculated on the basis of susceptible biotype named B 17/37 on a factor basis. Equation (response curve) has been now provided for each biotype. The equation used for estimating response is as:

y = [Intercept + bx (covariate x are transformed using the base 10.0 logarithm)]

Figure 2: Are those GR50 values or Resistant/susceptible factor significantly different from each other? If yes, What statistical test did you perform to find this out? Please explain in the figure legend.

In the revised version, we have now calculated LD50 value and these values were compared using confidence interval overlap as described in table 3. The probit analysis detail of each biotype has been provided in the supplementary file (SPSS software) .

I don’t see any depository for the raw data on these experiments. Please provide all your raw data from where you build these tables and figures. This is the most important requirement of the journal.

We have now provided all raw data on these experiments as supplementary file .

Reviewer #2: The topic is important. The whole paper is clear and well- structured. The experiment is reasonably designed. The result is mostly clear and fully described but I personally think that the result is probably not very well-dug. More analysis about the correlations among variables, the effect comparison of glyphosate usage could be conducted.

1. Please make sure with the journal whether the material and method parts are at the end of the paper or after introduction.

Yes, the material and method parts are at the end of the paper as per journal style.

2. I might miss some information, but it is better having more descriptions about the codes of the species, for example, B17/34, does it have some relationship with the sampling (Fig.1)? It is not so clear and very easy to get confused with these codes when reading.

For B17/34 indicates B is for biotype; 17 indicates sample was collected in 2017 and 34 is the serial number in collection. We reported the biotypes as per our designation in the weed library.

3. The Fig.1 seems not so clear and short of legend and north arrow. The color you chose is too similar with each other which makes the whole figure looks not so good.

The figure 1 has been modified with clear legend and north arrow.

4. There are correlations among leaf, tiller, amount of seeds, etc. Tables are good, but if you have several scatter plots to show these relationships, it might be clearer.

Many correlation between different parameters were significant, therefore, to avoid the confusion, we preferred to show this presentation in table form. 

5. For the influences of height, leaf, etc. to the final tiller of the species, there might be interactive effects among different x variables. Is it better you use the multi- correlation analysis, i.e., y = ax1+bx2+cx3…? This is just a suggestion and you could think about this.

We have correlation matrix in table 2 and critical differences were observed at 5% level of significance. 

 6. Line 134~135, it is not clear using “target” here.

We have replaced this with ‘restrict high’

7. I am not sure, for the unit, you use number plant-1 etc., and do you think plant-1 here is proper? Is it better just delete it and make some explanations in the text? Because all unit is measured by one individual. Or you just use stem-1or individual-1?

It is the standard trend that we have followed and easy to understand. 

8. In table 2, delete the last number “1”.

Deleted

9. For table 3, make clear the meaning of each column at the title. “Probit analysis detail” is not clear.

Added. However, all detail of probit analysis has been provided in supplementary file.

10. Line 275, “by shaking” is not precise, you might find another word to describe this operation. 

We feel that shaking best describes our method and have used this word in our previous papers to describe this.

11. Line 305~306, more explanation might be needed for the “Fisher’s protected LSD”? At least give the full name of LSD when it is firstly mentioned.

Full name of LSD (least significant differences) has been mentioned now.

12. Is the different doses of the glyphosate used well compared? It is not clear in the result.

LD50 values have now been calculated for each biotype and the values compared with a 95% confidence interval.

13. For reference 6 and 7, there are italic styles, make sure with journal and be consist with the others.

Italics have been removed.

14. You declared that the data is available in the supporting information but I did not see any materials attached, please explain this.

We have attached all raw data now.

Reviewer #3: Two experiments were carried out in this study to compare the growth behavior and glyphosate resistance among 10 biotypes of Echinochloa colona in Australia. The growth experiment showed that characteristics like tall nature and high tillering capacity allow E. colona to produce a high leaf number that result in a large number of seed heads and seeds. The glyphosate dose-response experiment showed a wide range of GR50 value in these biotypes, and the biotype B17/34 showed the strongest resistance to glyphosate. The two experiments were conducted rigorously with appropriate replication and sample sizes, and the conclusions were drawn appropriately based on the data presented. In addition, the language of this manuscript is well written and clear.

Thank you for your comments.

The analysis of variance (ANOVA) and Fisher’s protected LSD were applied to test the difference of growth behavior of E. colona among 10 biotypes. And the probit anaylsis were applied to estimate the GR50 value. These statistical methods were performed appropriately, but the outcome of these statistical results have not been sufficiently presented. 

Moreover, there are several major and specific comments on this manuscript:

1. The introduction can be reduced to 3 or 4 paragraphs to make the background and aim clearer. The outline of the introduction can refer to:

1) The Echinochloa colona as a noxious weed emerged in Australia crops.

2) The biological characteristics of growth and reproduction of E. colona.

3) The glyphosate is widely used in Australia to eliminate the E. colona, but induced a variety of glyphosate resistant biotypes.

4）the aim and scientific questions of this study

All your points have been addressed and the introduction has been reduced to 4 paragraphs. Differences among growth behaviour of biotypes were compared with LSD values.

2. The author combined the “results” and “discussion” in the manuscript. Although this format can be accepted by some journals, but I suggest divided this part into “results” and “discussions” separately. In the present manuscript, the result of two experiment were discussed respectively, and I can not find their relationships. If there is an independent section of discussion, the relationships between the two experiment and their implications can be better integrated.

Results and discussion are combined here according to the format of the journal and we feel combining the results and discussion avoids repetition. The relationship between the two experiments and the implications has been discussed in general discussion at the end of the article (Conclusion section).

3. The author proposed “growth behavior and seed production potential in these biotypes had no correlation with the resistance index” (LINE 240-241). However, the statistical methods for analyzing the correlation between growth traits and resistance index were not given in the section of “statistical analysis”. Also, I can not find the relevant result which support this conclusion in the context.

Statistical methods for analysing the correlation has been provided under heading statistical analysis. The relationships between parameters were assessed using linear regression analysis. Suggested lines have been added in the conclusion section. The analysis detail of correlation has been provided in the supplementary file. 

4. The seed of 10 biotypes of E. colona were collected from a large area in northern grain region in Australia. So, do the geographical differences effect on the growth behavior and glyphosate resistance among 10 biotypes of E. colona. I suggest the multivariate analysis such as PCA or RDA can be applied to study whether the biotypes from closer sites have similar characteristics. Moreover, the correlation (or non-correlation) between growth behavior and resistance index may also be revealed by the multivariate analysis.

We tested these biotypes under one location. Therefore, it is not justified to run PCA analysis. There was no such trend that close sites have similar characteristics. We are studying the genetic diversity of these biotypes in a different experiment. 

Here are also several specific comments:

5. Figure 1. It’s better to give a small-scale map of Australia, and pointed out the location of northern grain region.

We have modified figure 1 to incorporate these suggestions. 

6. Table 1. the standard deviation of these variables should be given. And as the LSD was performed, I suggest the author add lowercase superscript letters after the values to indicate significance at the 0.05 level for the differences among biotypes.

Letters are to be used in DMRT (Duncan multiple range test) design. LSD values have been given to identify the differences between biotypes. If the difference between two values is more than LSD value, it means the treatments are significantly different. However, we have included the standard deviation in the table.

7. LINE 64: “shikmate pathway” might be “shikimate pathway”

Corrected.

8. Line 251-254: it better to give more information on the seed collected location or the farm, like the crop type, soil type, glyphosate dose and so on, which can help readers know the growth environment of these biotypes.

We have provided the GPS coordinates on map.

However, we are including all information here for your information

Biotype Coordinates Location/Place Crop

B17/7 27.50000000/151.69666667 Dalby Wheat-Fallow

B17/12 30.268508/149.80481 Narrabri Wheat- Fallow

B17/13 30.306499/149.811438 Narrabri Wheat-Fallow

B17/16 30.09099349/149.64890 Narrabri Lathyrus-wheat/chickpea

B17/17 30.38230/149.59679 Narrabri Fallow

B17/25 28.31500000/148.68916667 St George Canal side

B17/34 28.58305556/150.36888889 Moree Wheat fallow

B17/35 29.95805/149.8339 Moree Cotton fallow

B17/37 27.5514/152.3428 Gatton Wheat-Fallow

B17/49 27.5514/152.3428 Gatton Wheat-Fallow

9. Line 252: how to identify different biotypes? Do they just collected from different sites，or have distinct heritable characteristics?

They were collected from different sites. Heritable characteristics were not studied in this experiment.

10. Line 301: I did not find the ANOVA result

ANOVA results have been provided as annexure.

Table 1. ANOVA for different parameters

Source Degree of freedom Plant height (cm) Tiller (number plant-1) Leaf (number plant-1) Seed head (number plant-1) Seed head weight (g plant-1 ) Shoot biomass (g plant-1) Root biomass (g plant-1) Days to seed head initiation (d) Seed production (number plant-1)

 Mean square

Replication 3 59.9 52.2 702.9 129.3 0.179 5.78 5.863 8.91 1621914

Experimental run 1 17155.2* 9812.4* 64139.5* 7940.1* 202.1* 15401.2* 315.96* 214.5* 0.433155*

Biotype 9 202.8* 323.5* 4340.5* 582.4* 12.5* 39.761* 50.14 43.0* 32654110*

Experimental run x Biotype 9 48.8 60.9 962.2 119.9 2.59 24.083 47.84 22.5 1001201

Error 57 32.0 44.1 925.5 200.9 2.75 14.584 24.5 7.34 5510769

* indicates significant at 5% level

11. Line 307-308 Which variable did the “GR50” in this study based on? The shoot biomass, tiller number, or individuals?

It was based on biomass. However, reviewer# 1 suggested to do on survival % basis. Therefore, now we have provided on survival % basis.

6. PLOS authors have the option to publish the peer review history of their article (what does this mean?). If published, this will include your full peer review and any attached files.

Yes

Do you want your identity to be public for this peer review? For information about this choice, including consent withdrawal, please see our Privacy Policy.

Reviewer #1: No

Reviewer #2: Yes: Liping Li

Reviewer #3: No

We are okay with Reviewer#2.

---

## [Decision Letter · Decision Letter 1]

2 Dec 2019

PONE-D-19-21773R1

Growth behavior and glyphosate resistance level in 10 biotypes of Echinochloa colona in Australia

PLOS ONE

Dear Dr Mahajan,

Thank you for submitting your manuscript to PLOS ONE. After careful consideration, we feel that it has merit but does not fully meet PLOS ONE’s publication criteria as it currently stands. Therefore, we invite you to submit a revised version of the manuscript that addresses the points raised during the review process.

ACADEMIC EDITOR: 

The manuscript has been greatly improved. However, there is still plenty of questions to answer and comments to address, as has been pointed out by reviewer #3. Besides, reviewer #1 asked that “Is there any plan to relate these phenotypic response back to management, cropping systems, location, and/or soils?” in the comments. But I did not find any response to this comment. Please response to this point in the next version. 

We would appreciate receiving your revised manuscript by Jan 16 2020 11:59PM. To enhance the reproducibility of your results, we recommend that if applicable you deposit your laboratory protocols in protocols.io, where a protocol can be assigned its own identifier (DOI) such that it can be cited independently in the future. For instructions see: http://journals.plos.org/plosone/s/submission-guidelines#loc-laboratory-protocols

We look forward to receiving your revised manuscript.

Kind regards,

Xiao Guo, Ph.D.

Academic Editor

PLOS ONE

Reviewers' comments:

Reviewer's Responses to Questions

**Comments to the Author**

1. If the authors have adequately addressed your comments raised in a previous round of review and you feel that this manuscript is now acceptable for publication, you may indicate that here to bypass the “Comments to the Author” section, enter your conflict of interest statement in the “Confidential to Editor” section, and submit your "Accept" recommendation.

Reviewer #2: All comments have been addressed

Reviewer #3: (No Response)

2. Is the manuscript technically sound, and do the data support the conclusions?

Reviewer #2: Yes

Reviewer #3: Yes

3. Has the statistical analysis been performed appropriately and rigorously? 

Reviewer #2: Yes

Reviewer #3: No

4. Have the authors made all data underlying the findings in their manuscript fully available?

Reviewer #2: Yes

Reviewer #3: Yes

5. Is the manuscript presented in an intelligible fashion and written in standard English?

Reviewer #2: Yes

Reviewer #3: Yes

6. Review Comments to the Author

Reviewer #2: Personally, I currently do not work a lot on spv files. So I think the supplemental materials are better in xlsx, or txt format. And it is better to have a list of the supporting information.

Reviewer #3: The manuscript PONE-D-19-21773 has been improved since the previous version. However, several problems have not yet been solved in the manuscript, although some of them were replied in the response letter. For example, I can see the ANOVA table and site information in the response letter, but can not find them neither in the manuscript nor in the supporting files. Moreover, the ANOVA statistics for the growth characteristics need to be reanalyzed, because the replication was set as a treatment, which I can not agree with. Another problem is, I think there needs some evidence or reference to prove that, the seeds being collected from one site are belong to the same biotype.

1. Line 248-250: By definition, the “biotypes” is a group of organisms having the same specific genotype. So, plants with same biotype must have common heritable traits, by which the biotype can be distinguished. In this study, the seeds of one biotype were collected from 40-50 plants per site and over an area >1ha. I wonder whether the seeds of same biotype were collected from one clone, or distinguished by any trait during the seed collection. Otherwise they can only be call “population” but not “biotype”. Since the experimental evidence for heritability traits is lacking, it is better to provide references in discussing this point.

2. Line 296, “14 January 2018” should be “14 January 2019”

3. Line 312-314, the author supplied the ANOVA table in the response letter, but I can not find it neither in the manuscript nor in the supporting material in xlsx files. Maybe it was included in the SPV format files which I could not open. The SPV file can only be open by SPSS, which is a commercial software and very expensive. I believe that many readers, just like me, have not installed the SPSS software. So, I suggested the contents in SPV files can be transformed into a common format, such as xlsx or txt, which can be opened by all readers.

Moreover, in the ANOVA table, the replicates should not be treated as a factor.

4. Line 319, remove “the growth of”

5. Figure 1, the author gave a table of the site information in the response letter, but I can neither find it in the manuscript nor supporting information.

6. Table 2, please give the sample size involved in the correlation analysis given in the table caption or note, like “n=?”

7. Figure 2 I found the biotypes of B17/7, B17/12, B17/17, B17/25 had no significant difference with the susceptible biotype. Do this mean the five biotypes being susceptible as B17/37? If it is, I suggest separate the all 10 biotypes into two groups, and compare their growth traits between groups. Maybe you can find significant differences of growth characteristics between susceptible and glyphosate resistant biotypes. This is just a suggestion and you could think about this.

8. Line 328: Furthermore, do the difference of LD50 and resistance index among the 10 biotypes were compared using LSD method? Please add the information of statistical methods .

9. Line 334, The “funding acquisition” can be removed because there was “no special funding was obtained for this work” in the funding information.

7. PLOS authors have the option to publish the peer review history of their article (what does this mean?). If published, this will include your full peer review and any attached files.

Reviewer #2: No

Reviewer #3: No

---

## [Author Response · Author response to Decision Letter 1]

8 Dec 2019

Dr Xiao Guo,

Academic Editor

PLOS ONE

Subject: Ref.: PONE-D-19-21773R1 Growth behavior and glyphosate resistance level in 10 biotypes of Echinochloa colona in Australia

PLOS ONE

Dear Dr Guo,

We thank you again for the critical look on the manuscript and highly appreciate the remarks of the reviewers for further improvement of the manuscript. The manuscript has been revised in the light of useful comments. All the comments have been incorporated in the revised version carefully. For clarity, the comments and suggestions will appear in the black colored text; while our response will appear in the blue text. In the manuscript also, specific changes can be seen in the highlighted text. We hope that the revised manuscript is now acceptable in “PLOS ONE”.

We look forward to hearing from you in due course.

Sincerely yours

Gulshan Mahajan

ACADEMIC EDITOR:

The manuscript has been greatly improved. However, there is still plenty of questions to answer and comments to address, as has been pointed out by reviewer #3. Besides, reviewer #1 asked that “Is there any plan to relate these phenotypic response back to management, cropping systems, location, and/or soils?” in the comments. But I did not find any response to this comment. Please response to this point in the next version. 

This study suggested that glyphosate resistant populations such as B17/34 that produced a high seed number are very problematic. Attempt should be made to reduce its invasion/further infestation. Movement of seeds from one field to another field via various mode of transportation should be restricted. For management of resistant population, crop competition tactics (narrow row spacing, competitive cultivars, optimum seeding rate, cover crops etc.) could be explored in an integrated weed management program to reduce the seed number of resistant populations. The related information has been added in the manuscript.

 Reviewer #2: Personally, I currently do not work a lot on spv files. So I think the supplemental materials are better in xlsx, or txt format. And it is better to have a list of the supporting information.

Agreed. The supplemental material of probit analysis has been attached in a Word file now.

Reviewer #3: The manuscript PONE-D-19-21773 has been improved since the previous version. However, several problems have not yet been solved in the manuscript, although some of them were replied in the response letter. For example, I can see the ANOVA table and site information in the response letter, but can not find them neither in the manuscript nor in the supporting files. Moreover, the ANOVA statistics for the growth characteristics need to be reanalyzed, because the replication was set as a treatment, which I can not agree with. Another problem is, I think there needs some evidence or reference to prove that, the seeds being collected from one site are belong to the same biotype.

The data have been again reanalysed with eight replications as per suggestions. An ANOVA table and site information has been added in the manuscript. The term biotype has been replaced with population. 

1. Line 248-250: By definition, the “biotypes” is a group of organisms having the same specific genotype. So, plants with same biotype must have common heritable traits, by which the biotype can be distinguished. In this study, the seeds of one biotype were collected from 40-50 plants per site and over an area >1ha. I wonder whether the seeds of same biotype were collected from one clone, or distinguished by any trait during the seed collection. Otherwise they can only be call “population” but not “biotype”. Since the experimental evidence for heritability traits is lacking, it is better to provide references in discussing this point.

As seeds were not collected from one clone. Therefore, to avoid confusion, we have replaced the term biotype with population.

2. Line 296, “14 January 2018” should be “14 January 2019”

Corrected.

3. Line 312-314, the author supplied the ANOVA table in the response letter, but I can not find it neither in the manuscript nor in the supporting material in xlsx files. Maybe it was included in the SPV format files which I could not open. The SPV file can only be open by SPSS, which is a commercial software and very expensive. I believe that many readers, just like me, have not installed the SPSS software. So, I suggested the contents in SPV files can be transformed into a common format, such as xlsx or txt, which can be opened by all readers.

Moreover, in the ANOVA table, the replicates should not be treated as a factor.

Now we have transferred the content from SPV file to txt file (probit analysis) and the file has been attached as a supplementary file. We have reanalysed the data again with eight replications. An ANOVA table has been added in the text.

4. Line 319, remove “the growth of”

Corrected

5. Figure 1, the author gave a table of the site information in the response letter, but I can neither find it in the manuscript nor supporting information.

Site information from Figure 1 has been added in the manuscript.

6. Table 2, please give the sample size involved in the correlation analysis given in the table caption or note, like “n=?”

Added. n=10.

7. Figure 2 I found the biotypes of B17/7, B17/12, B17/17, B17/25 had no significant difference with the susceptible biotype. Do this mean the five biotypes being susceptible as B17/37? If it is, I suggest separate the all 10 biotypes into two groups, and compare their growth traits between groups. Maybe you can find significant differences of growth characteristics between susceptible and glyphosate resistant biotypes. This is just a suggestion and you could think about this.

Yes, populations B17/7, B17/12, B17/17, B17/25 were susceptible as 17/37. However, they differ in their growth behaviour. We tried to analyse with your suggestion and found these cannot be grouped together on the basis of growth traits.

8. Line 328: Furthermore, do the difference of LD50 and resistance index among the 10 biotypes were compared using LSD method? Please add the information of statistical methods .

No, these were not compared using LSD method. LD50 values for each population were compared using confidence interval overlap in probit analysis (Table 4). Resistance index is a factor (resistance/susceptibility ratio); therefore, these were also compared on the basis of LD50 value.

9. Line 334, The “funding acquisition” can be removed because there was “no special funding was obtained for this work” in the funding information.

Yes, it has been removed.

---

## [Decision Letter · Decision Letter 2]

30 Dec 2019

Growth behavior and glyphosate resistance level in 10 biotypes of Echinochloa colona in Australia

PONE-D-19-21773R2

Dear Dr. Mahajan,

We are pleased to inform you that your manuscript has been judged scientifically suitable for publication and will be formally accepted for publication once it complies with all outstanding technical requirements.

With kind regards,

Xiao Guo, Ph.D.

Academic Editor

PLOS ONE

Additional Editor Comments (optional):

Reviewers' comments:

Reviewer's Responses to Questions

**Comments to the Author**

1. If the authors have adequately addressed your comments raised in a previous round of review and you feel that this manuscript is now acceptable for publication, you may indicate that here to bypass the “Comments to the Author” section, enter your conflict of interest statement in the “Confidential to Editor” section, and submit your "Accept" recommendation.

Reviewer #2: All comments have been addressed

Reviewer #3: All comments have been addressed

2. Is the manuscript technically sound, and do the data support the conclusions?

Reviewer #2: Yes

Reviewer #3: (No Response)

3. Has the statistical analysis been performed appropriately and rigorously? 

Reviewer #2: Yes

Reviewer #3: (No Response)

4. Have the authors made all data underlying the findings in their manuscript fully available?

Reviewer #2: Yes

Reviewer #3: (No Response)

5. Is the manuscript presented in an intelligible fashion and written in standard English?

Reviewer #2: Yes

Reviewer #3: (No Response)

6. Review Comments to the Author

Reviewer #2: The file “Probit analysis detail” looks strange to me. And is there a supplement material list? I did not see.

Reviewer #3: (No Response)

7. PLOS authors have the option to publish the peer review history of their article (what does this mean?). If published, this will include your full peer review and any attached files.

Reviewer #2: No

Reviewer #3: No

---

## [Editor Report · Acceptance letter]

6 Jan 2020

PONE-D-19-21773R2 

Growth behavior and glyphosate resistance level in 10 populations of *Echinochloa colona* in Australia 

Dear Dr. Mahajan:

I am pleased to inform you that your manuscript has been deemed suitable for publication in PLOS ONE. Congratulations! Your manuscript is now with our production department. 

With kind regards,

on behalf of

Dr. Xiao Guo 

Academic Editor

PLOS ONE